# FASTER TRAINING OF WORD EMBEDDINGS

## ABSTRACT

Word embeddings have gained increasing popularity in the recent years due to the Word2vec library and its extension fastText that uses subword information. In this paper, we aim at improving the execution speed of fastText training on homogeneous multi- and manycore CPUs while maintaining accuracy. We present a novel open-source implementation that flexibly incorporates various algorithmic variants including negative sample sharing, batched updates, and a byte-pair encoding-based alternative for subword units. We build these novel variants over a fastText implementation that we carefully optimized for the architecture, memory hierarchy, and parallelism of current manycore CPUs. Our experiments on three languages demonstrate 3–20× speed-up in training time at competitive semantic and syntactic accuracy.

## 1 INTRODUCTION

Word embeddings have a long history (Rumelhart et al., 1986; Bengio et al., 2003; Collobert & Weston, 2008), but have received much attention in recent years due to word2vec (Mikolov et al., 2013) and its computationally efficient implementation via skip-gram with negative sampling. Word embeddings capture contextual relationships between the words, and have become a standard input representation for the majority of NLP tasks, benefitting, e.g., classification (Joulin et al., 2016; Deriu et al., 2017) or machine translation (Jansen, 2017; Conneau et al., 2017). More recently, state-of-the-art results on many language understanding tasks were achieved by deep transformer architectures such as BERT (Devlin et al., 2019), which however are very compute intensive both at training and inference time, even with pre-trained models and reduced parameter space. Thus, simpler and more lightweight static word embeddings such as fastText (Bojanowski et al., 2017) are still widely used, due to their fast execution, comparable results for particular tasks (Tseng et al., 2019), and ability to produce a single vector per word, which helps in information retrieval with interpretability and search index construction.

**Contributions.** In this paper, we present algorithmic and code optimization techniques to improve the training time of word2vec and fastText embeddings on modern general-purpose multicore and manycore computers. We present an optimized open-source implementation of word2vec and fastText that encapsulates a number of algorithmic variants including negative sample sharing, batched updates, and subword units based on byte-pair encoding approach. Our extensive evaluation on three languages shows that the best combinations of optimizations speed up training time by 2.7–20.6 times while maintaining accuracy of selected NLP tasks.

## 2 WORD EMBEDDINGS

**Word2vec.** Word2vec is built upon a simple bilinear regression model trained on word co-occurrence, resulting in numerical feature representations as floating point vectors of dimensionality $d$. Given a word in a sentence, the goal of the algorithm is to maximize the likelihood of predicting surrounding (context) words. To achieve this, the model is trained to increase the probability of predicting particular words if they appear close to a given current word in the training corpus. A popular variant also decreases the probability of predicting words that do not appear close to the current word (negative sampling (Mikolov et al., 2013; Goldberg & Levy, 2014)).

During training, the algorithm processes the corpus in a streaming fashion. Each word $w_i$ (called current word) is processed together with its surrounding context words

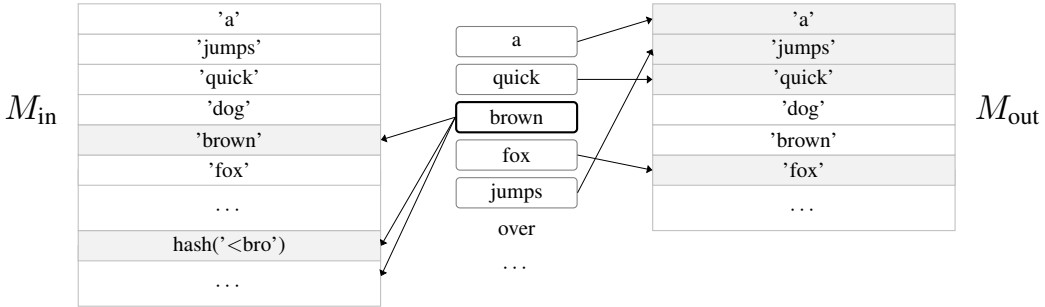

Figure 1: Representation of source and target words in the input ($M_{\text{in}}$) and output ($M_{\text{out}}$) matrix in fastText (skip-gram). The sentence is "a quick brown fox jumps over the lazy dog", the current word $w_i$ is "brown" and the context window size is 2. The words in the corpus are represented as indices of the corresponding rows in $M_{\text{in}}$ and $M_{\text{out}}$.

$\{w_{i-C}, ..., w_{i-1}\}, \{w_{i+1}, ..., w_{i+C}\}$, where $C$ is the range of the context window. There are two modes of operation training the model for the following prediction tasks:

- *Skip-gram (SG)*: predict target context words using the current word $w_i$ as the source.
- *CBOW*: predict the target current word $w_i$ using context words as the source.

Each word $w$ in the vocabulary of size $V$ is represented as a source $w_s$ by one row in the $V \times d$ *input* matrix $M_{\text{in}}$ containing word embeddings, and each word is represented as a target $w_t$ by one row in the $V \times d$ *output* matrix $M_{\text{out}}$ that is used to calculate the training objective function. The goal of the optimization is to maximize the inner products (minimize difference) of real pairs of source current words with the target context words or *vice versa*, using the binary logistic loss. This approach can be improved by the use of negative sampling, where the algorithm additionally maximizes the difference between the source current words and the words picked randomly out of the source's context. Training is performed using stochastic gradient descent (SGD). SGD is performed in parallel with $p$ threads by splitting the training corpus into $p$ parts and processing them asynchronously ("Hogwild" approach (Recht et al., 2011)). The final embedding of each word is its corresponding row in $M_{\text{in}}$. $M_{\text{out}}$ is discarded at the end of training.

**FastText.** FastText (Bojanowski et al., 2017) improves word2vec by utilizing subwords of target words during the training. A typical run of fastText uses subwords of lengths $k = 3 \dots 6$, containing delimiters $\langle, \rangle$ at the boundaries. For example, for the word *paris* and $k = 3$ the subwords are: $\langle$pa, par, ari, ris, is$\rangle$.

In fastText, the embeddings $M_{\text{in}}$ are extended to contain rows representing both entire words as well as hashes of all their subwords. Additionally, the representation of the entire word is added to the set of its subwords. During the execution of the algorithm, the hidden layer $\mathbf{h}$ is built by averaging vectors in $M_{\text{in}}$ representing the source word's subwords. The final vector embedding for each word is obtained in the same way. $M_{\text{out}}$ remains unchanged. Fig. 1 shows an example of how the word vectors are stored and accessed. A single update is described in Alg. 1.

**Related work.** FastText has been implemented as a part of the popular Gensim library (Rehurek & Sojka, 2011) using Cython and a default machine's BLAS library (e.g., Intel MKL) for algebraic operations. In our experiments we found the code memory-expensive and slow: training 5 epochs on a 1 GB English Wikipedia dump with 24 threads took approximately 11 hours on a Knights Landing CPU, about 10 times slower than the original fastText. Therefore, we use the original code provided by Facebook Research (2016a) as the baseline in all our experiments.

For skip-gram with negative sampling, pWord2Vec (Ji et al., 2016) transforms the "Hogwild" approach into "Hogbatch", by performing updates on multiple context words at once (effectively turning a series of dot products into a matrix-matrix operation) and sharing negative samples for the entire batch. We employ similar techniques in our implementation.

The work by Rengasamy et al. (2017) extends this approach by context combining, where multiple contexts can share a set of negative samples and be updated all at once. We do not adapt this approach as it requires careful preprocessing rewarded by a relatively small speedup over pWord2Vec.

---

**Algorithm 1:** A single iteration of the original fastText algorithm. In skip-gram, it is performed on each current-context word pair (as source-target). In CBOW, all context words are used as source words at the same time.

---

**Data:** source word(s) $w_s$, target word $w_t$, learning rate $l$, number of negative samples $n$

1    **if** *skip-gram* **then**            // **Initialize.**

2      $\mathbf{h} = \frac{M_{\text{in}}(w_s) + \sum_{z \in \text{subwords}(w_s)} M_{\text{in}}(z)}{\text{count}(\text{subwords}(w_s)) + 1}$    // Average vectors of the source word(s) and their

3    **else if** *CBOW* **then**          // subwords to obtain the hidden layer.

4      $\mathbf{h} = \frac{\sum_s (M_{\text{in}}(w_s) + \sum_{z \in \text{subwords}(w_s)} M_{\text{in}}(z))}{\sum_s (\text{count}(\text{subwords}(w_s)) + 1)}$

5    $\mathbf{g} = 0$              // Reset the gradient.

                                // **Update the target word.**

6    $\alpha = l(1 - \sigma(\mathbf{h} \cdot M_{\text{out}}(w_t)))$    // Compute positive score reflecting similarity between

                                //     $\mathbf{h}$ and the row $M_{\text{out}}(w_t)$ representing $w_t$.

7    $\mathbf{g} = \mathbf{g} + \alpha \cdot M_{\text{out}}(w_t)$    // Build the gradient.

8    $M_{\text{out}}(w_t) = M_{\text{out}}(w_t) + \alpha \cdot \mathbf{h}$    // Update the target word.

9    **for** $t' \leftarrow 1$ **to** $n$ **do**        // **Update negative samples: negative score.**

10      pick a random negative sample $w_{t'} \neq w_t$

11      $\alpha = l(0 - \sigma(\mathbf{h} \cdot M_{\text{out}}(w_{t'})))$    // Compute negative score.

12      $\mathbf{g} = \mathbf{g} + \alpha \cdot M_{\text{out}}(w_{t'})$    // Build the gradient.

13      $M_{\text{out}}(w_{t'})) = M_{\text{out}}(w_{t'})) + \alpha \cdot \mathbf{h}$    // Update the target word.

14    **end**

15    **if** *skip-gram* **then**         // **Update the source rows(s).**

16      $M_{\text{in}}(w_s) = M_{\text{in}}(w_s) + \mathbf{g}$

17      **foreach** $z \in subwords(w_s)$ **do**

18        $M_{\text{in}}(z) = M_{\text{in}}(z) + \mathbf{g}$

19      **end**

20    **else if** *CBOW* **then**

21      **foreach** $w_s$ **do**        // As a result, the difference between rows in $M_{\text{in}}(w_s)$

22        $M_{\text{in}}(w_s) = M_{\text{in}}(w_s) + \mathbf{g}$    // and $M_{\text{out}}$ which corresponds to the positive samples

23        **foreach** $z \in subwords(w_s)$ **do**    // drops, and the difference between rows in $M_{\text{out}}$

24          $M_{\text{in}}(z) = M_{\text{in}}(z) + \mathbf{g}$    // which correspond to the negative samples increases.

25        **end**

26      **end**

---

Word2vec and fastText have been also implemented for GPU clusters. BlazingText (Gupta & Khare, 2017) tackles the problem of efficient batch size and synchronization for multiple GPUs. While this issue is of no concern on CPU, they report the execution time on a single GPU comparable to a 16-threaded CPU fastText baseline. We further speed up the CPU implementation. The work by Bae & Yi (2016) reports up to $11\times$ speedup of word2vec with negative sampling run on a K20 GPU over the single-threaded CPU word2vec. However, they report only up to $1.6\times$ speedup over a 12-threaded CPU run. Our no_subword version of the code are roughly 5 (skip-gram) and 6 (CBOW) faster than the 20-threaded runs of the original word2vec. Word2vec and fastText are memory-intensive algorithms. Additionally, fine-grained parallelism is limited by relatively small vectors typically used in the computations. These characteristics severely limit the potential advantages of GPU over CPU.

Li et al. (2019) discuss a distributed version for many GPUs aiming at the reduction of write conflicts in updates. Similarly (and independently), we made attempts at pre-scheduling a list of current-context word updates, but we found the overhead of this preprocessing prohibitive. Nonetheless, the algorithmic variants presented in our paper can be applied in distributed setting, as long as the data used for a single update fits inside a batch used in the distributed computation. This will be the case for our variants, since they either execute separate updates on each current-context word pair, or update current words with their entire (typically small) contexts. This is also the case in the original fastText and therefore, the communication cost should not increase.

Another popular word embedding model is GloVe (Pennington et al., 2014). While the Authors claim superiority over word2vec, a more thorough evaluation (e. g., by Wang et al. (2019), or Kumar et al. (2020)) shows that there is no clear winner, as the results may vary depending on the training corpus, evaluation task and hyperparameters used. Additionally, GloVe lacks the information on word morphology provided by fastText, and does not scale well for large vocabularies. Since GloVe

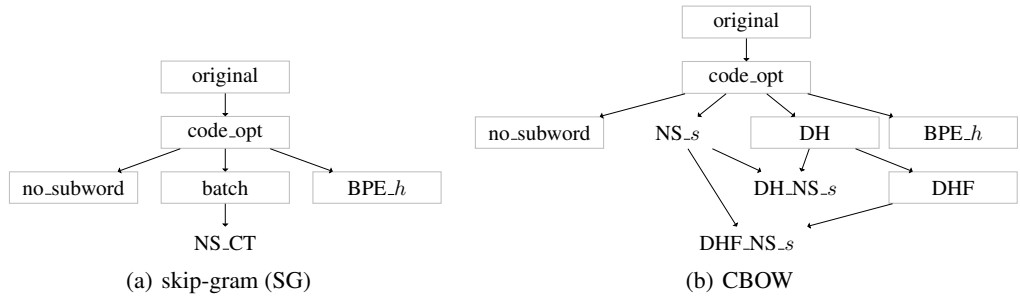

Figure 2: Dependency between our code variants of skip-gram and CBOW. The experiments for the "NS" variants (no box frames) are only shown in the appendix due to inferior experimental accuracy.

is based on a completely different algorithmic structure, that is, creation and reduction of a global word co-occurrence matrix, there is no straightforward way to apply to it our code optimizations and variants.

## 3 OPTIMIZATION TECHNIQUES AND ALGORITHMIC VARIANTS

To improve the training time, we first identify the most expensive operations. Assume that the source word(s) have a total of $m$ subwords and that we use $n$ negative samples per target word. Then each update comprises:

- Construction of $\mathbf{h}$: a sum of $m$ vectors (line 2 or 4).
- Loss calculation: $n + 1$ dot products and $2(n + 1)$ vector additions (lines 6–8, 11–13).
- Gradient update: $m$ vector additions (lines 16–19 or 21–26).

In skip-gram, $\mathbf{h}$ is built only from a single current word, while in CBOW, it is constructed from all context words. The loss function, in contrast, is computed once per each current-context word pair in skip-gram, while in CBOW, the loss is computed using the entire context. This means that for CBOW, the construction of $\mathbf{h}$ and the gradient update consumes most of the execution time, while for skip-gram, these operations take roughly the same amount of time as the loss function calculation (assuming the default parameter of $n = 5$). All operations listed above are memory-intensive and therefore memory bound: thus, the best approach to optimize them is by reducing memory movement and avoiding unnecessary updates. It is further supported by our observations during the tests on isolated sections of the code. We noted that a large amount of execution time is taken by the latency of lower levels of memory hierarchy when accessing data from the rows scattered across $M_{\text{in}}$ and $M_{\text{out}}$, which is an access pattern required to provide high quality embeddings.

To speed up the training, we first perform a number of code performance optimizations and then various algorithmic modifications compared to the original fastText. Some modifications depend on each other as illustrated in Fig. 2. Some, but not all, techniques apply to both modes of operation. All our improvements build on code_opt which is a CPU-specific optimization of the original fastText code. For skip-gram, we consider a batch variant and negative sharing across the context (NS_CT). For CBOW, we consider keeping track of the values in the hidden layer $\mathbf{h}$ and updating them dynamically rather than building this layer from scratch in each iteration (DH: variable context window size, DHF: fixed context window size). We consider combinations of this technique with negative sharing involving different number $s$ of positive samples that the negative samples are shared between (NS_s). Additionally, for both CBOW and skip-gram, we test no_subword where we remove the subwords from code_opt, making it equivalent to optimized word2vec, and BPE_h, where we replace samples with BPE tokens obtained from a pre-trained token set of size $h$.

We next discuss these variants, referring to specific parts of Alg. 1 that they modify.

**Code performance optimizations (code_opt).** For efficient execution we explicitly vectorize matrix and vector operations using AVX-512 intrinsics. We block and merge operations involving multiple reads from the same location in memory to make them more cache-friendly (temporal locality), like averaging the rows of $M_{\text{in}}$ or subsequently reading from and writing to $M_{\text{out}}$.

During the creation of hidden layer $\mathbf{h}$ (line 2 or 4), we reduce the number of array accesses such that each element of $\mathbf{h}$ is stored only once during summing up the vector representations of subwords. This improves the original code which performs a separate store for each subword.

To speed up the binary logistic loss function, we vectorize the dot product (lines 6, 11) with the use of eight accumulators to increase instruction-level parallelism without too much register pressure. We merge the update of the gradient $\mathbf{g}$ and the relevant rows of $M_{\text{out}}$ (lines 7–8, 12–13) to avoid multiple reads from the latter. We still call the loss function once per each $w_t$ and $w_t'$.

Similar to the creation of $\mathbf{h}$, we improve the update of $M_{\text{in}}(w_s)$ with $\mathbf{g}$ (line 16 or 21–22) by reading each element of $\mathbf{g}$ only once for all words and subwords $w_s$.

The optimizations in the version code_opt are used in all algorithmic variants discussed next. Note that these optimizations can also be applied to other regularization schemes such as the hierarchical softmax used in the original word2vec (Mikolov et al., 2013).

**No subwords (no_subword).** Experiments in Section 4 show that it is sometimes useful to train word embeddings without any subword information. We provide a code variant which disables subwords, but applies all optimizations discussed above. It is algorithmically equivalent to word2vec. In Alg. 1, $subwords(w_s)$ in lines 1–4 and 15–26 becomes an empty set and needs not be processed. While this is expected to improve the training time, especially for CBOw which dedicated a large part of its runtime to averaging and updating subword representations, the information on the word morphology becomes scarce. We will later see that the word embeddings trained without subword information do not perform well when used for syntactic tasks. On the other hand, the training then focuses on semantic information which is reflected in higher semantic quality of these embeddings.

**Minibatching (batch for SG).** For skip-gram, we implement a form of minibatching of the target words per each source word. Rather than following the work of Ji et al. (2016), which merges all $M_{\text{in}}(w_s)$ rows in a minibatch into a matrix, we follow the original fastText's approach hitherto only applied to CBOW, which simply averages all these rows. The advantage of our minibatching over the original fastText skip-gram is being able to execute a single update for each context window of the current word $w_i$, rather than per each current-context word pair. This means that $\mathbf{h}$ and the relevant rows of the input matrix $M_{\text{in}}$ are updated only once per each current word, independent of context window size. Lines 2 and 16 are now executed only once per context window, in a similar fashion as in lines 4 and 21–22 respectively). This creates an additional delay between reading and writing a word's subword representations increasing the possibility of write conflicts, but our experiments later show that the accuracy remains nearly unaffected. Minibatching can bring significant speed improvements to subword-based training due to the relatively high cost of building $\mathbf{h}$ and updating all subword representations. As mentioned, in fastText CBOW, this form of batching is already a part of its algorithmic structure.

**Negative sharing (NS_CT and NS_s).** We implement negative sharing proposed by Ji et al. (2016), but adapted for and built over $SG\_batch$ and the natural batching of fastText CBOW. For skip-gram, we share negative samples among all words in the entire context window of $w_i$ (NS_CT). For CBOW, we share negative samples for $s$ consecutive current words $w_i$ (NS_s). In our implementation, $s$ is a hyperparameter chosen by the user. Thus, line 10 is executed only $n$ times every $s$-th update. While negative sharing results in fewer expensive random memory reads and improved memory locality (e. g., for $d = 300$, $n = 5$, and a context window size of 11, the data worked upon takes up c. a. 16 KB, while an usual L1 cache size is not less than 32 KB), it proportionally reduces the number of data samples used in training per current-context word pair. For this reason, despite improvements in execution time, NS yields inferior accuracy. Therefore, we do not report its results in the paper, but only present them in the appendix.

**Dynamic hidden layer update (DH).** CBOW spends a large portion of its execution time building $\mathbf{h}$ and updating relevant rows of $M_{\text{in}}$ for each subsequent current word $w_i$ and its context window. Therefore, we opt for adding and removing subwords only as their words move in and out of the context window as the algorithm processes the training text. After each shift of the context window, we update the rows of $M_{\text{in}}$ for all removed subwords, readjust to the gradient $\mathbf{g}$, and add new subwords to $\mathbf{h}$. Thus, rather than performing the entire sum in line 4, the data is processed in five steps. Assuming that $x$ embeddings remain inside the context window after a particular shift:

1. Denormalize $\mathbf{h}$.

2. Update $M_{\text{in}}$ for subwords falling out of the context window.
3. Subtract embeddings of subwords falling out of the context window from $\mathbf{h}$.
4. Readjust to gradient $\mathbf{g}$: $\mathbf{h} = \mathbf{h} + x\mathbf{g}$.
5. Add subwords falling into the context window to $\mathbf{h}$.
6. Normalize $\mathbf{h}$.

Note that this creates additional delay between reading and writing to the rows of $M_{\text{in}}$, but empirically it does not harm the vector quality.

**Fixed window for dynamic hidden layer update (DHF).** Since the window size is picked randomly in each iteration, some words will fall in and out of the context window multiple times, forcing DH to remove and add the same subwords to $\mathbf{h}$ multiple times over a short period of time. To mitigate this, we fix the window size. While potentially saving time, this approach comes with a pitfall: the variable window size is a natural way of sampling context words that are closer to a current word $w_i$ with greater probability, which reflects a greater contribution of these words to the current word's meaning. DHF effectively ignores the impact of the distance of context words.

**Byte-Pair vocabulary (BPE_$h$).** We also propose an alternative approach to subword embeddings, replacing the subwords by Byte-Pair Encoding (BPE) tokens (Sennrich et al., 2016). These are produced with the Hugging Face Tokenizers library (Moi, 2019) in the form of token IDs for the $h$ most frequent word fragments, where $h$ is a hyperparameter. We expect this to reduce execution time and memory consumption as the number of tokens is typically an order of magnitude smaller than that of subwords. To our knowledge, this is the first attempt to apply BPE tokenization to provide additional subword information in fastText-like fashion. In our experiments in Section 4, we train the tokenizer over the same training corpus as our embeddings, but both trainings could use different corpora. In case the BPE variant of fastText is unable to tokenize a word found in its training corpus (e.g., because it was absent from the corpus used for training the tokenizer), the word remains as it is, without additional embeddings. An alternative approach would be to create embeddings for tokens consisting of single characters: however, we found that if many words fail to be tokenized, this may cause a drastic slowdown, likely due to update conflicts on the single-character tokens. In Alg. 1, using the BPE variant means replacing "subwords" with "tokens".

## 4 EVALUATION

In this section we evaluate our performance-optimized polyalgorithmic implementation for training word embeddings on a current homogeneous multicore system. For each algorithmic variant, and considering three languages, we report the speedup we achieve for training and the obtained accuracy of the generated word embeddings w.r.t. a number of semantic and syntactic quality tests.

**Setup.** We use a dual-socket Intel(R) Xeon(R) Silver 4114 CPU processor (Skylake, 20 physical cores). For evaluation, we create an English corpus as described by Facebook Research (2016b). For other languages, we proceed in analogous fashion: download respective Wikipedia dumps (Wikimedia Foundation, 2001), sanitize and lowercase with the script wikifil.pl authored by Mahoney (2006). For each language, the script is modified to capture relevant characters and replace relevant words. We truncate the outputs to 1 billion characters. The resulting vocabulary sizes are: (a) 218,316 words for English, (b) 592,674 words for German, (c) 385,596 words for Russian. The purpose of our experiments is to speed up training over the original fastText. We demonstrate the speed-ups of our implementation and show which algorithmic variants maintain accuracy at the same time.

The results for English are presented in Table 1. The names of algorithmic variants match those from Section 3 and Fig. 2. We omit negative sharing (NS) due to low accuracy, but show the results in Appendix F. In tokenized runs (BPE), we use $h = 20\text{K}, 40\text{K}, 200\text{K}$. All other hyperparameters are the fastText defaults. The speedups shown are over the original implementations SG_original and CBOW_original, respectively, from Bojanowski et al. (2017), run with the same number of threads. The scaling column shows the speedup of our code when run with 20 threads compared to 1 thread. The runtimes are consistent over several runs and are shown in Appendix F.

We perform various semantic and syntactic accuracy tests explained below. The best accuracy scores for each test are marked in blue. Pareto-optimal combinations of accuracy scores are shown bold-faced. Pareto-optimal means that no other algorithmic variant dominates it, i.e., is better on each

Table 1: Accuracy and speedup achieved with our library over fastText when training on English Wikipedia corpus. Blue: best accuracy in category, bold: Pareto-optimal accuracy, speedup is over the original fastText run with the same number of threads, and scaling is the speedup of 20 threads vs. 1 thread for our code. Higher is better for all metrics.

| algorithmic variant | accuracy | | | | | | speedup (times) | | scaling |
| | QW | | BATS | | MUSE | Battig | 1 thread | 20 threads | |
| | sem. | syn. | sem. | syn. | | | | | |
| SG_original | **27.07** | **64.22** | **9.30** | **41.54** | 0.643 | **40.97** | 1.0 | 1.0 | 15.2 |
| *Our work:* | | | | | | | | | |
| SG_code_opt | **26.37** | **63.75** | **9.29** | **41.61** | 0.647 | **40.85** | 3.7 | 2.7 | 11.2 |
| SG_no_subword | **47.42** | **48.07** | **12.97** | **27.81** | 0.635 | **41.27** | 10.2 | 8.8 | 13.1 |
| SG_batch | **24.28** | **62.47** | **8.92** | **41.84** | 0.630 | **40.76** | 4.4 | 3.6 | 12.2 |
| SG_BPE_20K | **40.78** | **53.85** | **12.86** | **34.50** | 0.642 | **40.43** | 4.9 | 4.2 | 13.0 |
| SG_BPE_40K | **47.77** | **50.56** | **13.33** | **33.16** | 0.646 | **39.80** | 4.8 | 4.2 | 13.2 |
| SG_BPE_200K | 54.98 | **44.79** | **12.65** | **27.73** | 0.634 | 41.46 | 4.8 | 4.2 | 13.4 |
| CBOW_original | 9.27 | 67.74 | 5.53 | 63.42 | 0.546 | 32.98 | 1.0 | 1.0 | 13.2 |
| *Our work:* | | | | | | | | | |
| CBOW_code_opt | 9.55 | 68.14 | 5.69 | 63.52 | 0.538 | 33.07 | 4.6 | 2.3 | 6.6 |
| CBOW_no_subword | **51.00** | **54.49** | 14.84 | **32.29** | 0.614 | 40.68 | 20.6 | 20.4 | 13.1 |
| CBOW_DH | **11.38** | 68.73 | **6.51** | 63.58 | 0.583 | 36.30 | 4.8 | 2.7 | 7.4 |
| CBOW_DHF | **5.29** | **57.44** | **3.94** | **51.71** | 0.633 | 31.37 | 4.8 | 3.3 | 9.2 |
| CBOW_BPE_20K | **22.65** | **46.01** | **10.17** | **36.98** | 0.607 | 35.56 | 11.4 | 11.0 | 12.8 |
| CBOW_BPE_40K | 30.85 | 40.47 | 10.13 | 33.38 | 0.614 | 36.07 | 11.5 | 11.1 | 12.8 |
| CBOW_BPE_200K | 43.33 | 26.40 | 8.12 | 24.70 | 0.606 | 40.39 | 11.4 | 11.3 | 13.0 |

score. Together with Fig. 2, the tables also show the incremental impact on accuracy and execution speed of each variant.

**Accuracy tests.** We perform multiple evaluation tasks to test the quality of our embeddings. First, we test our embeddings with the word analogy task script provided with word2vec (Mikolov et al., 2013) for both *semantic* and *syntactic* accuracy. For English, we use the questions-words (QW) dataset (Mikolov, 2013). For other languages, we use its translations: German (Köper et al., 2015) and Russian (Kononova, 2017). Additionally, we employ the Vecto library (Vecto, 2018) to evaluate on the The Bigger Analogy Test Set (BATS) (Gladkova et al., 2016) on English embeddings, with the 3CosAdd method. For all analogy benchmarks, we observe that fastText performs better for syntactic than semantic tasks as was already noted by Bojanowski et al. (2017). Second, we compute word similarity scores with Facebook MUSE (Conneau et al., 2017), using monolingual evaluation with word similarity tasks on semantic datasets. For English and German, we use the tests provided by MUSE. For Russian, we use the HJ dataset (Panchenko et al., 2016). The tables in this section present the averaged MUSE output. For detailed score for each MUSE test set, see Appendix B. Third, we use the scripts provided by the Word Embedding Benchmarks package (Jastrzebski, 2015) to perform the concept categorization (word clustering) task. We evaluate on the semantic Battig test set introduced by Battig & Montague (1969).

**Evaluating English skip-gram.** First, we evaluate multiple variants of skip-gram presented in the first section of Tab. 1. The dependencies between the variants is illustrated in Fig. 2(a). We observe that only optimizing for efficient execution (code_opt) yields for fastText a 2.7–3.7× speedup while maintaining accuracy. The no_subword variant yields 8–10× speedup over original and about 3× speedup over code_opt. For word analogy, no_subword improves the embeddings semantically. The tokenized versions roughly balance between the fastText- and word2vec-style embedding quality, with an exception of BPE_200K where the number of tokens is close to the vocabulary size, effectively turning only the most common subwords into separate tokens. This approach provides a semantic accuracy even greater than original for both BATS and QW, however at a price of syntactic quality, all including roughly 4–5× speedup over original and up to 1.5× speedup over code_opt. For QW, the accuracies vary greatly, while BATS indicates that a smaller number of tokens is generally preferable. The batch variant maintains or slightly handicaps the accuracy of fastText, and provides a slightly smaller speedup than the tokenized versions. The different variants of skip-gram

perform almost equally well on the word similarity and categorization tasks, and all of them yield Pareto-optimal results. All variants show good parallel scaling.

**Evaluating English CBOW.** The CBOW results are shown in the second section of Tab. 1; the dependencies between variants are in Fig. 2(b). The code_opt variant yields 2.3–4.6× speedup over original, less than for skip-gram, but the obtained accuracy is not Pareto-optimal. For word analogy, CBOW generally performs better on syntactic than semantic questions. The no_subword variant provides good scaling, and over 20× speedup over original and about 8–9× speedup over code_opt. It diminishes the discrepancy between these scores, albeit impacting negatively the syntactic quality of the embeddings, while achieving highest scores for word similarity and categorization. None of the tokenized variants was able to beat no_subword both in speed and evaluation on these tasks, but they provide an improvement in semantic accuracy over original, as well as in word similarity and categorization. The BPE variants achieve roughly 11× speedup over original. The DH variant provides only a slight speedup over code_opt (2.7–4.8× speedup over original), but yields higher accuracies in all tasks, while DHF impacts negatively all scores except for MUSE, but provides a speedup over DH with multiple threads.

**Comparison between skip-gram and CBOW.** As a rule of thumb, the fastText implementations of skip-gram perform much better on semantic questions in word analogy tasks and slightly better in word similarity and categorization tasks. For syntactic questions, the CBOW_code_opt and CBOW_DH variants are a better option. On the other hand, CBOW_no_subword performs nearly as well for word similarity and categorization tasks as skip-gram. Therefore, in specific cases, the former can be used in lieu of skip-gram to boost the execution speed.

**Evaluation on German and Russian corpora.** Table 2 contains results for German and Russian, presented analogously to those in English. In terms of evaluation accuracy, they are largely consistent with English, with the small exception of CBOW_BPE_20K, which performs better than CBOW_no_subword on the German corpus. This indicates the impact of the number of tokens used during training and opens opportunities for further investigation. Noteworthy, CBOW_DH achieves the best scores on syntactic tasks for all evaluated languages. Using skip-gram with BPE tokens rather than fastText-style subwords performs very well in terms of both speedups and accuracy scores, all of which are Pareto-optimal. The code_opt variants yields slightly better speedups than for English, and further optimizations lead to significantly greater speedups. The code_opt variants yield roughly 3.5–5× speedup over their respective original versions. The best achieved improvement is CBOW_no_subword, up to 50× for Russian. This shows that our improvements are particularly beneficial for morphologically-rich languages with a large number of subwords per word.

## 5 CONCLUSIONS

We presented a thorough evaluation, and associated open-source implementation, of various optimization techniques for fastText and word2vec. In particular, these include code-level performance optimizations and the use of BPE tokens rather than subwords. For example, for English, our code offers practitioners speedups in the range of 2.7–20.6×, while maintaining a single or multi-dimensional notion of accuracy. We achieve good parallel scaling, which is expected to bring even more benefits in the future, as the number of cores further increases. The choice of algorithm depends heavily on the accuracy metric: for all languages, there is no universally best variant, which makes a case for our polyalgorithmic implementation and thorough evaluation of trade-offs. Our techniques should also apply to sent2vec (Pagliardini et al., 2018) for sentence embeddings.

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

Table 2: Accuracy and speedup achieved with our library over fastText when training on (a) German and (b) Russian Wikipedia corpora. Blue: best accuracy in category, bold: Pareto-optimal accuracy, "speedup": over the original fastText run with the same number of threads, "scaling": speedup 20 threads vs. 1 thread for our code. Higher is better for all metrics.

| algorithmic variant | accuracy | | | speedup (times) | | scaling |
|---|---|---|---|---|---|---|
| | QW | | MUSE | 1 | 20 | |
| | sem. | syn. | | thread | threads | |
| SG_original | **21.13** | **49.94** | **0.587** | 1.0 | 1.0 | 8.5 |
| *Our work:* | | | | | | |
| SG_code_opt | 19.41 | 49.12 | 0.575 | 3.5 | 4.9 | 11.8 |
| SG_no_subword | **42.31** | **27.29** | **0.589** | 11.5 | 18.8 | 13.9 |
| SG_batch | 17.43 | 49.13 | 0.573 | 4.6 | 7.0 | 12.9 |
| SG_BPE_20K | **29.91** | **36.83** | **0.589** | 5.5 | 8.9 | 13.9 |
| SG_BPE_40K | **34.55** | **29.96** | **0.593** | 5.2 | 9.0 | 14.6 |
| SG_BPE_200K | **45.90** | **23.39** | **0.593** | 5.3 | 9.0 | 14.6 |
| CBOW_original | 4.71 | 58.94 | 0.507 | 1.0 | 1.0 | 8.3 |
| *Our work:* | | | | | | |
| CBOW_code_opt | 4.30 | 59.09 | 0.510 | 4.3 | 3.4 | 6.7 |
| CBOW_no_subword | **37.75** | **27.66** | **0.559** | 24.1 | 41.8 | 14.3 |
| CBOW_DH | **5.96** | **59.81** | **0.527** | 4.9 | 4.2 | 7.0 |
| CBOW_DHF | **1.86** | **54.17** | **0.591** | 5.2 | 4.5 | 7.2 |
| CBOW_BPE_20K | 11.19 | 31.31 | 0.542 | 13.1 | 21.1 | 13.3 |
| CBOW_BPE_40K | 18.67 | 23.25 | 0.557 | 13.3 | 22.0 | 13.7 |
| CBOW_BPE_200K | 27.81 | 16.14 | 0.560 | 12.9 | 22.6 | 14.5 |

(a) German

| algorithmic variant | accuracy | | | speedup (times) | | scaling |
|---|---|---|---|---|---|---|
| | QW | | MUSE | 1 | 20 | |
| | sem. | syn. | | thread | threads | |
| SG_original | **12.29** | **77.61** | **0.633** | 1.0 | 1.0 | 10.2 |
| *Our work:* | | | | | | |
| SG_code_opt | **12.86** | **77.81** | **0.622** | 4.1 | 4.2 | 10.4 |
| SG_no_subword | **23.82** | **42.89** | **0.588** | 12.1 | 16.5 | 14.0 |
| SG_batch | 10.58 | 76.32 | 0.629 | 5.1 | 5.2 | 10.5 |
| SG_BPE_20K | **14.22** | **51.86** | **0.621** | 5.8 | 7.9 | 13.8 |
| SG_BPE_40K | **16.75** | **46.81** | **0.608** | 5.8 | 7.9 | 13.9 |
| SG_BPE_200K | **22.90** | **45.91** | **0.600** | 5.8 | 8.0 | 14.1 |
| CBOW_original | 8.88 | 80.78 | 0.495 | 1.0 | 1.0 | 7.9 |
| *Our work:* | | | | | | |
| CBOW_code_opt | **9.18** | **79.79** | **0.497** | 4.5 | 4.6 | 8.0 |
| CBOW_no_subword | 16.01 | 37.35 | 0.532 | 28.1 | 50.5 | 14.2 |
| CBOW_DH | **9.17** | **81.43** | **0.523** | 4.3 | 5.7 | 10.5 |
| CBOW_DHF | **7.32** | **78.75** | **0.557** | 4.6 | 6.2 | 10.6 |
| CBOW_BPE_20K | 8.28 | 34.82 | 0.513 | 15.0 | 25.2 | 13.3 |
| CBOW_BPE_40K | 8.96 | 40.66 | 0.493 | 15.2 | 26.1 | 13.6 |
| CBOW_BPE_200K | 11.13 | 35.96 | 0.537 | 15.3 | 27.0 | 14.0 |

(b) Russian

William F Battig and William E Montague. Category norms of verbal items in 56 categories a replication and extension of the connecticut category norms. *Journal of experimental Psychology*, 80(3p2):1, 1969.

Yoshua Bengio, Réjean Ducharme, Pascal Vincent, and Christian Jauvin. A neural probabilistic language model. *Journal of machine learning research*, 3(Feb):1137–1155, 2003.

Piotr Bojanowski, Edouard Grave, Armand Joulin, and Tomas Mikolov. Enriching word vectors with subword information. *Transactions of the Association for Computational Linguistics*, 5:135–146, 2017. doi: 10.1162/tacl_a_00051. URL `https://www.aclweb.org/anthology/Q17-1010`.

Ronan Collobert and Jason Weston. A unified architecture for natural language processing: Deep neural networks with multitask learning. In *Proceedings of the 25th international conference on Machine learning*, pp. 160–167, New York, NY, USA, 2008. Association for Computing Machinery. doi: 10.1145/1390156.1390177.

Alexis Conneau, Guillaume Lample, Marc'Aurelio Ranzato, Ludovic Denoyer, and Hervé Jégou. Word translation without parallel data. *arXiv preprint arXiv:1710.04087*, 2017.

Jan Deriu, Aurelien Lucchi, Valeria De Luca, Aliaksei Severyn, Simon Müller, Mark Cieliebak, Thomas Hofmann, and Martin Jaggi. Leveraging large amounts of weakly supervised data for multi-language sentiment classification. In *Proceedings of the 26th International Conference on World Wide Web*, pp. 1045–1052, Republic and Canton of Geneva, CHE, 2017. International World Wide Web Conferences Steering Committee. doi: 10.1145/3038912.3052611. URL `https://doi.org/10.1145/3038912.3052611`.

Jacob Devlin, Ming-Wei Chang, Kenton Lee, and Kristina Toutanova. BERT: Pre-training of deep bidirectional transformers for language understanding. pp. 4171–4186, 2019. doi: 10.18653/v1/N19-1423. URL `https://www.aclweb.org/anthology/N19-1423`.

Facebook Research. *facebookresearch/fastText: Library for fast text representation and classification.*, 2016a. `https://github.com/facebookresearch/fastText`.

Facebook Research. *Word representations*, 2016b. `https://fasttext.cc/docs/en/unsupervised-tutorial.html`.

Anna Gladkova, Aleksandr Drozd, and Satoshi Matsuoka. Analogy-based detection of morphological and semantic relations with word embeddings: what works and what doesn't. In *Proceedings of the NAACL Student Research Workshop*, pp. 8–15, 2016.

Yoav Goldberg and Omer Levy. word2vec explained: deriving mikolov et al.'s negative-sampling word-embedding method. *arXiv preprint arXiv:1402.3722*, 2014.

Saurabh Gupta and Vineet Khare. Blazingtext: Scaling and accelerating word2vec using multiple gpus. In *Proceedings of the Machine Learning on HPC Environments*, pp. 1–5. 2017.

Stefan Jansen. Word and phrase translation with word2vec. *arXiv preprint arXiv:1705.03127*, 2017.

Stanislaw Jastrzebski. *kudkudak/word-embeddings-benchmarks: Package for evaluating word embeddings.*, 2015. `https://github.com/kudkudak/word-embeddings-benchmarks`.

Shihao Ji, Nadathur Satish, Sheng Li, and Pradeep Dubey. Parallelizing word2vec in multi-core and many-core architectures. *arXiv preprint arXiv:1611.06172*, 2016.

Armand Joulin, Edouard Grave, Piotr Bojanowski, Matthijs Douze, Hérve Jégou, and Tomas Mikolov. Fasttext.zip: Compressing text classification models. *arXiv preprint arXiv:1612.03651*, 2016.

Tatyana Kononova. *Russian adaptation of Google Analogies Dataset*, 2017. `https://rusvectores.org/static/testsets/ru_analogy_tagged_PROPN.txt`.

Maximilian Köper, Christian Scheible, and Sabine Schulte im Walde. Multilingual reliability and "semantic" structure of continuous word spaces. In *Proceedings of the 11th international conference on computational semantics*, pp. 40–45, London, UK, 2015. Association for Computational Linguistics. URL `https://www.aclweb.org/anthology/W15-0105`.

P Santosh Kumar, Rakesh Bahadur Yadav, and Sunita Vikrant Dhavale. A comparison of pre-trained word embeddings for sentiment analysis using deep learning. In *International Conference on Innovative Computing and Communications*, pp. 525–537. Springer, 2020.

Bofang Li, Aleksandr Drozd, Yuhe Guo, Tao Liu, Satoshi Matsuoka, and Xiaoyong Du. Scaling word2vec on big corpus. *Data Science and Engineering*, 4(2):157–175, 2019.

Matt Mahoney. *About the Test Data*, 2006. https://mattmahoney.net/dc/textdata.html#appendixa.

Tomas Mikolov. *Questions-Words.TXT*, 2013. https://github.com/nicholas-leonard/word2vec/blob/master/questions-words.txt.

Tomas Mikolov, Kai Chen, Greg Corrado, and Jeffrey Dean. Efficient estimation of word representations in vector space. *arXiv preprint arXiv:1301.3781*, 2013.

Anthony Moi. *Fast State-of-the-Art Tokenizers optimized for Research and Production*, 2019. https://github.com/huggingface/tokenizers.

Matteo Pagliardini, Prakhar Gupta, and Martin Jaggi. Unsupervised learning of sentence embeddings using compositional n-gram features. In *Proceedings of the 2018 Conference of the North American Chapter of the Association for Computational Linguistics: Human Language Technologies, Volume 1 (Long Papers)*, pp. 528–540, New Orleans, Louisiana, 2018. Association for Computational Linguistics. doi: 10.18653/v1/N18-1049. URL https://www.aclweb.org/anthology/N18-1049.

Alexander Panchenko, Dmitry Ustalov, Nikolay Arefyev, Denis Paperno, Natalia Konstantinova, Natalia Loukachevitch, and Chris Biemann. Human and machine judgements for russian semantic relatedness. In *International conference on analysis of images, social networks and texts*, pp. 221–235. Springer, Cham, 2016. URL https://doi.org/10.1007/978-3-319-52920-2_21.

Jeffrey Pennington, Richard Socher, and Christopher D Manning. Glove: Global vectors for word representation. In *Proceedings of the 2014 conference on empirical methods in natural language processing (EMNLP)*, pp. 1532–1543, 2014.

Benjamin Recht, Christopher Re, Stephen Wright, and Feng Niu. Hogwild: A lock-free approach to parallelizing stochastic gradient descent. In *Advances in Neural Information Processing Systems 24*, pp. 693–701. Curran Associates, Inc., 2011. URL http://papers.nips.cc/paper/4390-hogwild-a-lock-free-approach-to-parallelizing-stochastic-gradient-descent.pdf.

Radim Rehurek and Petr Sojka. Gensim—statistical semantics in python. *Retrieved from genism.org*, 2011.

Vasudevan Rengasamy, Tao-Yang Fu, Wang-Chien Lee, and Kamesh Madduri. Optimizing word2vec performance on multicore systems. In *Proceedings of the Seventh Workshop on Irregular Applications: Architectures and Algorithms*, pp. 1–9, New York, NY, USA, 2017. Association for Computing Machinery. doi: 10.1145/3149704.3149768. URL https://doi.org/10.1145/3149704.3149768.

David E Rumelhart, Geoffrey E Hinton, and Ronald J Williams. Learning representations by back-propagating errors. *Nature*, 323(6088):533–536, 1986. URL https://doi.org/10.1038/323533a0.

Rico Sennrich, Barry Haddow, and Alexandra Birch. Neural machine translation of rare words with subword units. pp. 1715–1725, 2016. doi: 10.18653/v1/P16-1162. URL https://www.aclweb.org/anthology/P16-1162.

Hou-Chiang Tseng, Hsueh-Chih Chen, Kuo-En Chang, Yao-Ting Sung, and Berlin Chen. An innovative bert-based readability model. In *International Conference on Innovative Technologies and Learning*, pp. 301–308. Springer, Cham, 2019. URL https://doi.org/10.1007/978-3-030-35343-8_32.

Vecto. *vecto-ai/vecto: Doing things with embeddings.*, 2018. https://github.com/vecto-ai/vecto.

Bin Wang, Angela Wang, Fenxiao Chen, Yuncheng Wang, and C-C Jay Kuo. Evaluating word embedding models: Methods and experimental results. *APSIPA transactions on signal and information processing*, 8, 2019.

Wikimedia Foundation. *Wikimedia Downloads*, 2001. `https://dumps.wikimedia.org`.

## A    DETAILS OF EXPERIMENTAL SETUP

For the experiments with questions-words and MUSE, we do not remove any questions containing out-of-vocabulary words. We use full test data sets for all modes, therefore our results are consistent.

For MUSE, we use the following tests provided by the library[1].

English:

EN_YP-130
EN_SIMLEX-999
EN_MTurk-771
EN_RG-65
EN_VERB-143
EN_SEMEVAL17
EN_MTurk-287
EN_MC-30
EN_RW-STANFORD
EN_WS-353-SIM
EN-TR-3k
EN_WS-353-ALL
EN_WS-353-REL

German:

DE_ZG222
DE_GUR65
DE_SIMLEX-999
DE_GUR350
DE_SEMEVAL17
DE_WS-353

For the final MUSE scores, we take arithmetic mean of individual scores obtained from these tests.

For Russian, MUSE provides no tests. Therefore, we download and use the HJ dataset[2].

## B    DETAILED EVALUATION WITH MUSE

Tables 3 and 4 present detailed MUSE results for each test set. For Russian, we use only one test set, thus the results would be redundant with those in Section 4 and Appendix F. For the English and German tables, we run the experiments on embeddings obtained in a different training run: therefore, the individual scores are not expected to average exactly to those in Section 4.

## C    PREPARATION OF DATA

To prepare German and Russian Wikipedia dumps for training, we modify the wikifil.pl script such that it captures relevant characters and replaces all digits with relevant words in each language. For German, we add *äöüß* to the set of Latin characters. For Russian, we extract Cyrillic characters. Then, we manually truncate the parsed texts to 1 billion characters, and further truncate to the last complete word in the resulting text. For example, if the German enumeration was cut at the 1 billion boundary, such that

---

[1]https://dl.fbaipublicfiles.com/arrival/wordsim.tar.gz
[2]https://github.com/nlpub/russe-evaluation/blob/master/russe/evaluation/hj.csv

Table 3: Detailed results of MUSE tests for fastText when training on English Wikipedia corpus. Higher is better. The results are obtained from different training run, hence slight difference from the main results.

| algorithmic variant | accuracy | | | | | | | | | | | | |
|---|---|---|---|---|---|---|---|---|---|---|---|---|---|
| | YP -130 | SIMLEX -999 | MTurk -771 | RG -65 | VERB -143 | SEME VAL17 | MTurk -287 | MC -30 | RW-STA NFORD | WS-353 -SIM | TR -3k | WS-353 -ALL | WS-353 -REL |
| SG_original | 0.532 | 0.758 | 0.470 | 0.765 | 0.449 | 0.725 | 0.665 | 0.675 | 0.718 | 0.666 | 0.762 | 0.760 | 0.357 |
| *Our work:* | | | | | | | | | | | | | |
| SG_code_opt | 0.520 | 0.785 | 0.480 | 0.819 | 0.443 | 0.716 | 0.673 | 0.691 | 0.737 | 0.665 | 0.776 | 0.764 | 0.366 |
| SG_no_subword | 0.524 | 0.813 | 0.375 | 0.842 | 0.416 | 0.730 | 0.672 | 0.687 | 0.745 | 0.669 | 0.796 | 0.752 | 0.369 |
| SG_batch | 0.463 | 0.772 | 0.462 | 0.768 | 0.424 | 0.717 | 0.678 | 0.685 | 0.728 | 0.656 | 0.735 | 0.767 | 0.364 |
| SG_NS_CT | 0.450 | 0.739 | 0.448 | 0.660 | 0.481 | 0.685 | 0.662 | 0.636 | 0.680 | 0.622 | 0.659 | 0.735 | 0.338 |
| SG_BPE_20K | 0.468 | 0.793 | 0.409 | 0.842 | 0.431 | 0.711 | 0.658 | 0.692 | 0.749 | 0.671 | 0.795 | 0.750 | 0.376 |
| SG_BPE_40K | 0.547 | 0.814 | 0.404 | 0.810 | 0.411 | 0.721 | 0.658 | 0.706 | 0.760 | 0.675 | 0.754 | 0.759 | 0.365 |
| SG_BPE_200K | 0.499 | 0.788 | 0.373 | 0.823 | 0.429 | 0.712 | 0.672 | 0.686 | 0.739 | 0.673 | 0.783 | 0.756 | 0.358 |
| CBOW_original | 0.364 | 0.610 | 0.403 | 0.738 | 0.374 | 0.662 | 0.570 | 0.494 | 0.543 | 0.563 | 0.711 | 0.706 | 0.367 |
| *Our work:* | | | | | | | | | | | | | |
| CBOW_code_opt | 0.350 | 0.604 | 0.404 | 0.701 | 0.377 | 0.663 | 0.585 | 0.508 | 0.546 | 0.568 | 0.698 | 0.706 | 0.365 |
| CBOW_no_subword | 0.392 | 0.769 | 0.381 | 0.774 | 0.435 | 0.706 | 0.663 | 0.648 | 0.709 | 0.658 | 0.749 | 0.721 | 0.390 |
| CBOW_NS_11 | 0.368 | 0.607 | 0.391 | 0.682 | 0.379 | 0.669 | 0.594 | 0.513 | 0.547 | 0.572 | 0.681 | 0.706 | 0.353 |
| CBOW_NS_80 | 0.378 | 0.657 | 0.412 | 0.700 | 0.384 | 0.643 | 0.652 | 0.611 | 0.625 | 0.608 | 0.667 | 0.694 | 0.318 |
| CBOW_NS_160 | 0.369 | 0.683 | 0.408 | 0.632 | 0.386 | 0.641 | 0.656 | 0.628 | 0.645 | 0.598 | 0.590 | 0.667 | 0.293 |
| CBOW_DH | 0.426 | 0.677 | 0.416 | 0.731 | 0.383 | 0.691 | 0.653 | 0.565 | 0.617 | 0.616 | 0.715 | 0.731 | 0.358 |
| CBOW_DH_NS_11 | 0.425 | 0.640 | 0.395 | 0.742 | 0.364 | 0.668 | 0.640 | 0.539 | 0.583 | 0.603 | 0.699 | 0.718 | 0.340 |
| CBOW_DH_NS_80 | 0.344 | 0.703 | 0.418 | 0.709 | 0.364 | 0.648 | 0.678 | 0.658 | 0.667 | 0.632 | 0.627 | 0.710 | 0.319 |
| CBOW_DH_NS_160 | 0.147 | 0.479 | 0.248 | 0.446 | 0.284 | 0.440 | 0.446 | 0.460 | 0.463 | 0.360 | 0.462 | 0.469 | 0.212 |
| CBOW_DHF | 0.466 | 0.751 | 0.514 | 0.820 | 0.449 | 0.691 | 0.624 | 0.653 | 0.710 | 0.676 | 0.792 | 0.737 | 0.386 |
| CBOW_DHF_NS_11 | 0.452 | 0.753 | 0.481 | 0.791 | 0.345 | 0.666 | 0.653 | 0.704 | 0.726 | 0.647 | 0.740 | 0.731 | 0.345 |
| CBOW_DHF_NS_80 | 0.366 | 0.736 | 0.418 | 0.743 | 0.346 | 0.636 | 0.637 | 0.695 | 0.701 | 0.627 | 0.636 | 0.705 | 0.320 |
| CBOW_DHF_NS_160 | 0.276 | 0.669 | 0.398 | 0.704 | 0.350 | 0.567 | 0.601 | 0.673 | 0.670 | 0.579 | 0.599 | 0.664 | 0.283 |
| CBOW_BPE_20K | 0.35 | 0.767 | 0.408 | 0.808 | 0.432 | 0.689 | 0.668 | 0.650 | 0.714 | 0.648 | 0.734 | 0.712 | 0.367 |
| CBOW_BPE_40K | 0.354 | 0.787 | 0.373 | 0.818 | 0.439 | 0.702 | 0.661 | 0.664 | 0.727 | 0.649 | 0.711 | 0.712 | 0.372 |
| CBOW_BPE_200K | 0.385 | 0.758 | 0.360 | 0.738 | 0.414 | 0.716 | 0.672 | 0.641 | 0.704 | 0.655 | 0.724 | 0.719 | 0.393 |

```
eins zwei drei vier
```

becomes

```
eins zwei dr
```

we truncate it to

```
eins zwei
```

.

Finally, we use `iconv` to ensure UTF-8 format.

For example, for the Russian Wikipedia dump, the order of actions is:

```
wget <wiki dump address>/<ru.dump>
perl wikifil-ru.pl <ru.dump> > ruwiki
head -c 1000000000 ruwiki > ruwiki9
# manually truncate the text
# to the last complete word
iconv -t utf-8 ruwiki9 -o ruwiki9-utf
```

We train the embeddings on the file `ruwiki9-utf`.

## D COMPATIBILITY WITH HUGGING FACE TOKENIZERS

We implement tokenization in a way that it is compatible with the files produced by `ByteLevelBPETokenizer` in the Hugging Face Tokenizers library[3]. We apply the same char-

---

[3]https://github.com/huggingface/tokenizers

Table 4: Detailed results of MUSE tests for fastText when training on German Wikipedia corpus. Higher is better. The results are obtained from different training run, hence slight difference from the main results.

| algorithmic variant | accuracy | | | | | |
|---|---|---|---|---|---|---|
| | ZG222 | GUR65 | SIMLEX-999 | GUR350 | SEMEVAL17 | WS-353 |
| SG_original | 0.339 | 0.588 | 0.372 | 0.726 | 0.659 | 0.730 |
| *Our work:* | | | | | | |
| SG_code_opt | 0.373 | 0.617 | 0.371 | 0.724 | 0.699 | 0.716 |
| SG_no_subword | 0.395 | 0.639 | 0.363 | 0.718 | 0.677 | 0.796 |
| SG_batch | 0.372 | 0.627 | 0.418 | 0.733 | 0.700 | 0.747 |
| SG_NS_CT | 0.332 | 0.600 | 0.388 | 0.702 | 0.688 | 0.700 |
| SG_BPE_20K | 0.387 | 0.632 | 0.358 | 0.720 | 0.692 | 0.703 |
| SG_BPE_40K | 0.395 | 0.644 | 0.385 | 0.722 | 0.707 | 0.749 |
| SG_BPE_200K | 0.385 | 0.638 | 0.327 | 0.714 | 0.691 | 0.772 |
| CBOW_original | 0.284 | 0.517 | 0.308 | 0.695 | 0.570 | 0.701 |
| *Our work:* | | | | | | |
| CBOW_code_opt | 0.284 | 0.517 | 0.319 | 0.693 | 0.579 | 0.695 |
| CBOW_no_subword | 0.357 | 0.627 | 0.363 | 0.709 | 0.658 | 0.672 |
| CBOW_NS_11 | 0.310 | 0.525 | 0.302 | 0.690 | 0.596 | 0.690 |
| CBOW_NS_80 | 0.320 | 0.534 | 0.372 | 0.653 | 0.659 | 0.600 |
| CBOW_NS_160 | 0.307 | 0.480 | 0.356 | 0.608 | 0.610 | 0.511 |
| CBOW_DH | 0.313 | 0.543 | 0.344 | 0.709 | 0.618 | 0.672 |
| CBOW_DH_NS_11 | 0.324 | 0.541 | 0.322 | 0.690 | 0.624 | 0.687 |
| CBOW_DH_NS_80 | 0.332 | 0.546 | 0.388 | 0.631 | 0.678 | 0.561 |
| CBOW_DH_NS_160 | 0.282 | 0.469 | 0.291 | 0.601 | 0.590 | 0.510 |
| CBOW_DHF | 0.387 | 0.642 | 0.442 | 0.695 | 0.710 | 0.675 |
| CBOW_DHF_NS_11 | 0.376 | 0.647 | 0.417 | 0.664 | 0.724 | 0.699 |
| CBOW_DHF_NS_80 | 0.339 | 0.540 | 0.406 | 0.591 | 0.619 | 0.587 |
| CBOW_DHF_NS_160 | 0.267 | 0.441 | 0.276 | 0.572 | 0.524 | 0.496 |
| CBOW_BPE_20K | 0.319 | 0.620 | 0.361 | 0.689 | 0.660 | 0.622 |
| CBOW_BPE_40K | 0.338 | 0.628 | 0.388 | 0.709 | 0.650 | 0.607 |
| CBOW_BPE_200K | 0.343 | 0.649 | 0.332 | 0.713 | 0.660 | 0.684 |

acter mapping for UTF-8 characters that use more than one byte, and preprocess the words from the vocabulary such that each begins with a special delimiter character $\dot{G}$.

Note that the number of tokens $h$ must be selected **during** tokenization with the Tokenizers library.

```
bpe = ByteLevelBPETokenizer()
bpe.train([<corpus file>],
          vocab_size=<h>)
bpe.save(<path>, <filename>)
```

(Note the `[]` brackets.)

## E    HOW TO RUN EXPERIMENTS

We provide parameterized code [4] to replicate our experiments. It is a modification of the original fastText library[5]. Note that we disabled the production of .bin file to reduce saving time and save storage space. Our experiments apply to unsupervised training with skip-gram and CBOW.

In order to compile, CMake, Intel ICPC compiler and a CPU with AVX-512 support are required. Please compile with:

```
cmake .
make
```

---

[4] https://github.com/FT-Submit/ft-mod

[5] https://github.com/facebookresearch/fastText

To run, please use the command:

```
fasttext {cbow, skipgram} \
  -input  <corpus file> \
  -output <embeddings file> \
  <arguments>
```

**Selecting code optimizations.** To run a particular algorithm mode, use the argument `-mode <mode>`. Table 5 explains all available modes.

Table 5: An overview of `-mode` arguments and their connection to our experiments. Parameter $s$ is set with `-shared `.

| argument | mode |
|---|---|
| `-mode normal` | *_code_opt (default) |
| `-mode batched` | SG_batch (skip-gram only) |
| `-mode ns` | SG_NS_CT (skip-gram)
CBOW_NS_$s$ (CBOW) |
| `-mode dh` | CBOW_DH (CBOW only) |
| `-mode dhf` | CBOW_DHF (CBOW only) |
| `-mode dh-ns` | CBOW_DH_NS_$s$ (CBOW only) |
| `-mode dhf-ns` | CBOW_DHF_NS_$s$ (CBOW only) |

By default, $s$ (the number of words sharing negative samples) is set to $2C + 1$, where $C$ is the maximum context window size (set in the original fastText with the argument `-ws <c>`). To set a different $s$, use `-shared `. Setting this argument to zero will result in the default setting.

**Selecting BPE tokens or no_subword.** To use BPE tokens instead of subwords, provide paths to the merge and vocab files produced by the Tokenizers library.

```
-token-merges <path/to/f.txt >
-token-vocab  <path/to/f.json>
```

Both these arguments must be set to enable the BPE run. The number of tokens $h$ is obtained from the merge and vocab files. For our experiments, we produce these files using our training corpora. Note that these two arguments are incompatible with `-no-subwords`.

Finally, to run the no_subword (word2vec) version of our code, use the argument `-no-subwords`. Note that it is incompatible with `-token-merges` and `-token-vocab`.

We run both the BPE and no_subword experiments using `-mode normal` (default).

The remaining arguments are identical to those used by the original fastText code.

To obtain the results for SG_original and CBOW_original, please run the original library.

## F  COMPLETE RESULTS: ENGLISH, RUSSIAN AND GERMAN

We provide complete results for the embeddings trained on English (Tab. 6), German (Tab. 7) and Russian (Tab. 8) corpora. In negative sharing (NS), we use $s = 11, 80, 160$.

## G  VARIANCE OF RESULTS

Due to low stability of word embeddings across different trainings (e.g., (Antoniak & Mimno, 2018)), we perform multiple experiments on our variants to characterize standard deviations obtained for the scores of word analogy task on the questions-words test set (QW). We measure across

Table 6: Full results: accuracy, runtime and speedup achieved with our library over fastText when training on English Wikipedia corpus. Blue: best accuracy in category, bold: Pareto-optimal accuracy, speedup is over the original fastText run with the same number of threads, and scaling is the speedup of 20 threads vs. 1 thread for our code. For accuracy, speedup and scaling, higher is better. For time, lower is better.

| algorithmic variant | accuracy | | | | | | time (s) | | speedup (times) | | scaling |
| | QW | | BATS | | MUSE | Battig | 1 thread | 20 threads | 1 thread | 20 threads | |
| | sem. | syn. | sem. | syn. | | | | | | | |
| SG_original | **27.07** | **64.22** | **9.30** | **41.54** | **0.643** | **40.97** | 33640 | 2221 | 1.0 | 1.0 | 15.2 |
| *Our work:* | | | | | | | | | | | |
| SG_code_opt | **26.37** | **63.75** | **9.29** | **41.61** | **0.647** | **40.85** | 9087 | 810 | 3.7 | 2.7 | 11.2 |
| SG_no_subword | **47.42** | **48.07** | **12.97** | **27.81** | **0.635** | **41.27** | 3309 | 253 | 10.2 | 8.8 | 13.1 |
| SG_batch | **24.28** | **62.47** | **8.92** | **41.84** | **0.630** | **40.76** | 7631 | 625 | 4.4 | 3.6 | 12.2 |
| SG_NS_CT | **13.10** | **60.52** | **6.74** | **42.87** | **0.611** | **36.99** | 2537 | 257 | 13.3 | 8.7 | 9.9 |
| SG_BPE_20K | **40.78** | **53.85** | **12.86** | **34.50** | **0.642** | **40.43** | 6843 | 527 | 4.9 | 4.2 | 13.0 |
| SG_BPE_40K | **47.77** | **50.56** | **13.33** | **33.16** | **0.646** | **39.80** | 6948 | 525 | 4.8 | 4.2 | 13.2 |
| SG_BPE_200K | **54.98** | **44.79** | **12.65** | **27.73** | **0.634** | **41.46** | 7041 | 526 | 4.8 | 4.2 | 13.4 |
| CBOW_original | 9.27 | 67.74 | 5.53 | 63.42 | 0.546 | 32.98 | 19614 | 1484 | 1.0 | 1.0 | 13.2 |
| *Our work:* | | | | | | | | | | | |
| CBOW_code_opt | 9.55 | 68.14 | 5.69 | 63.52 | 0.538 | 33.07 | 4285 | 651 | 4.6 | 2.3 | 6.6 |
| CBOW_no_subword | 51.00 | 54.49 | **14.84** | **32.29** | 0.614 | 40.68 | 951 | 73 | 20.6 | 20.4 | 13.1 |
| CBOW_NS_11 | 7.18 | 63.30 | 4.96 | 60.27 | 0.545 | 32.63 | 3623 | 605 | 5.4 | 2.5 | 6.0 |
| CBOW_NS_80 | 6.11 | 63.41 | 4.70 | 60.27 | 0.570 | 33.74 | 3604 | 604 | 5.4 | 2.5 | 6.0 |
| CBOW_NS_160 | 5.10 | 62.50 | 4.41 | 60.85 | 0.555 | 30.74 | 3584 | 602 | 5.5 | 2.5 | 6.0 |
| CBOW_DH | **11.38** | **68.73** | **6.51** | **63.58** | 0.583 | 36.30 | 4079 | 555 | 4.8 | 2.7 | 7.4 |
| CBOW_DH_NS_11 | 9.19 | 64.19 | 6.03 | 58.77 | 0.564 | 34.66 | 3422 | 480 | 5.7 | 3.1 | 7.1 |
| CBOW_DH_NS_80 | 6.64 | 56.76 | 5.05 | 48.63 | 0.574 | 32.82 | 3347 | 474 | 5.9 | 3.1 | 7.1 |
| CBOW_DH_NS_160 | 3.90 | 54.84 | 3.77 | 49.19 | 0.536 | 29.44 | 3371 | 454 | 5.8 | 3.3 | 7.4 |
| CBOW_DHF | **5.29** | **57.44** | **3.94** | **51.71** | **0.633** | **31.37** | 4100 | 446 | 4.8 | 3.3 | 9.2 |
| CBOW_DHF_NS_11 | 2.71 | 44.51 | 2.75 | 43.38 | 0.608 | 29.94 | 3214 | 398 | 6.1 | 3.7 | 8.1 |
| CBOW_DHF_NS_80 | 2.84 | 47.19 | 3.10 | 43.26 | 0.582 | 29.65 | 3136 | 392 | 6.3 | 3.8 | 8.0 |
| CBOW_DHF_NS_160 | 2.00 | 52.77 | 2.63 | 53.79 | 0.539 | 24.78 | 3132 | 392 | 6.3 | 3.8 | 8.0 |
| CBOW_BPE_20K | 22.65 | 46.01 | **10.17** | **36.98** | **0.607** | 35.56 | 1717 | 135 | 11.4 | 11.0 | 12.8 |
| CBOW_BPE_40K | 30.85 | 40.47 | 10.13 | 33.38 | 0.614 | 36.07 | 1710 | 133 | 11.5 | 11.1 | 12.8 |
| CBOW_BPE_200K | 43.33 | 26.40 | 8.12 | 24.70 | 0.606 | 40.39 | 1714 | 131 | 11.4 | 11.3 | 13.0 |

5 runs. We were unable to perform more tests due to time constraints, but the results provide hints on stability. We present the results in Table 9. We note that the standard deviation rarely increases above 1. Moreover, it is more likely to be high for no_subword and BPE variants of the code.

Table 7: Full results: accuracy, runtime and speedup achieved with our library over fastText when training on German Wikipedia corpus. Blue: best accuracy in category, bold: Pareto-optimal accuracy, speedup is over the original fastText run with the same number of threads, and scaling is the speedup of 20 threads vs. 1 thread for our code. For accuracy, speedup and scaling, higher is better. For time, lower is better.

| algorithmic variant | accuracy | | | time (s) | | speedup (times) | | scaling |
|---|---|---|---|---|---|---|---|---|
| | QW | | MUSE | 1 thread | 20 threads | 1 thread | 20 threads | |
| | sem. | syn. | | | | | | |
| SG_original | **21.13** | **49.94** | **0.587** | 51747 | 6095 | 1.0 | 1.0 | 8.5 |
| *Our work:* | | | | | | | | |
| SG_code_opt | 19.41 | 49.12 | 0.575 | 14599 | 1233 | 3.5 | 4.9 | 11.8 |
| SG_no_subword | **42.31** | **27.29** | **0.589** | 4511 | 325 | 11.5 | 18.8 | 13.9 |
| SG_batch | 17.43 | 49.13 | 0.573 | 11249 | 870 | 4.6 | 7.0 | 12.9 |
| SG_NS_CT | 7.08 | 47.41 | 0.547 | 4506 | 383 | 11.5 | 15.9 | 11.8 |
| SG_BPE_20K | **29.91** | **36.83** | **0.589** | 9440 | 682 | 5. | 8.94 | 13.9 |
| SG_BPE_40K | **34.55** | **29.96** | **0.593** | 9908 | 678 | 5.2 | 9.0 | 14.6 |
| SG_BPE_200K | **45.90** | **23.39** | **0.593** | 9830 | 675 | 5.3 | 9.0 | 14.6 |
| CBOW_original | 4.71 | 58.94 | 0.507 | 31674 | 3837 | 1.0 | 1.0 | 8.3 |
| *Our work:* | | | | | | | | |
| CBOW_code_opt | 4.30 | 59.09 | 0.510 | 7452 | 1114 | 4.3 | 3.4 | 6.7 |
| CBOW_no_subword | 37.75 | **27.66** | **0.559** | 1315 | 92 | 24.1 | 41.8 | 14.3 |
| CBOW_NS_11 | 3.17 | 55.67 | 0.520 | 5754 | 1050 | 5.5 | 3.7 | 5.5 |
| CBOW_NS_80 | 3.04 | 52.50 | 0.515 | 5664 | 1068 | 5.6 | 3.6 | 5.3 |
| CBOW_NS_160 | 1.58 | 46.71 | 0.472 | 5615 | 1045 | 5.6 | 3.7 | 5.4 |
| CBOW_DH | **5.96** | **59.81** | **0.527** | 6406 | 911 | 4.9 | 4.2 | 7.0 |
| CBOW_DH_NS_11 | **4.57** | 55.56 | **0.530** | 5532 | 841 | 5.7 | 4.6 | 6.6 |
| CBOW_DH_NS_80 | 2.30 | 50.95 | 0.522 | 5812 | 831 | 5.5 | 4.6 | 7.0 |
| CBOW_DH_NS_160 | 1.51 | 45.25 | 0.450 | 6097 | 829 | 5.2 | 4.6 | 7.4 |
| CBOW_DHF | **1.86** | **54.17** | **0.591** | 6117 | 845 | 5.2 | 4.5 | 7.2 |
| CBOW_DHF_NS_11 | 0.82 | 46.36 | 0.562 | 5049 | 751 | 6.3 | 5.1 | 6.7 |
| CBOW_DHF_NS_80 | 1.03 | 46.97 | 0.498 | 5028 | 737 | 6.3 | 5.2 | 6.8 |
| CBOW_DHF_NS_160 | 1.08 | 41.07 | 0.420 | 5059 | 764 | 6.3 | 5.0 | 6.6 |
| CBOW_BPE_20K | 11.19 | 31.31 | 0.542 | 2413 | 182 | 13.1 | 21.1 | 13.3 |
| CBOW_BPE_40K | 18.67 | 23.25 | 0.557 | 2389 | 174 | 13.3 | 22.0 | 13.7 |
| CBOW_BPE_200K | 27.81 | 16.14 | 0.560 | 2461 | 170 | 12.9 | 22.6 | 14.5 |

Table 8: Full results: accuracy, runtime and speedup achieved with our library over fastText when training on Russian Wikipedia corpus. Blue: best accuracy in category, bold: Pareto-optimal accuracy, speedup is over the original fastText run with the same number of threads, and scaling is the speedup of 20 threads vs. 1 thread for our code. For accuracy, speedup and scaling, higher is better. For time, lower is better.

| algorithmic variant | accuracy | | | time (s) | | speedup (times) | | scaling |
|---|---|---|---|---|---|---|---|---|
| | QW | | MUSE | 1 | 20 | 1 | 20 | |
| | sem. | syn. | | thread | threads | thread | threads | |
| SG_original | **12.29** | **77.61** | **0.633** | 30219 | 2959 | 1.0 | 1.0 | 10.2 |
| *Our work:* | | | | | | | | |
| SG_code_opt | **12.86** | **77.81** | **0.622** | 7409 | 710 | 4.1 | 4.2 | 10.4 |
| SG_no_subword | **23.82** | 42.89 | 0.588 | 2501 | 179 | 12.1 | 16.5 | 14.0 |
| SG_batch | 10.58 | 76.32 | 0.629 | 5914 | 565 | 5.1 | 5.2 | 10.5 |
| SG_NS_CT | 5.58 | 77.81 | 0.602 | 2289 | 254 | 13.2 | 11.7 | 9.0 |
| SG_BPE_20K | **14.22** | **51.86** | **0.621** | 5179 | 376 | 5.8 | 7.9 | 13.8 |
| SG_BPE_40K | **16.75** | **46.81** | **0.608** | 5212 | 374 | 5.8 | 7.9 | 13.9 |
| SG_BPE_200K | **22.90** | **45.91** | **0.600** | 5230 | 370 | 5.8 | 8.0 | 14.1 |
| CBOW_original | 8.88 | 80.78 | 0.495 | 20510 | 2599 | 1.0 | 1.0 | 7.9 |
| *Our work:* | | | | | | | | |
| CBOW_code_opt | **9.18** | **79.79** | **0.497** | 4541 | 569 | 4.5 | 4.6 | 8.0 |
| CBOW_no_subword | 16.01 | 37.35 | 0.532 | 731 | 51 | 28.1 | 50.5 | 14.2 |
| CBOW_NS_11 | 7.92 | 77.32 | 0.493 | 4166 | 519 | 4.9 | 5.0 | 8.0 |
| CBOW_NS_80 | 7.46 | 75.83 | 0.490 | 4105 | 518 | 5.0 | 5.0 | 7.9 |
| CBOW_NS_160 | 6.61 | 68.75 | 0.391 | 4311 | 518 | 4.8 | 5.0 | 8.3 |
| CBOW_DH | **9.17** | **81.43** | **0.523** | 4790 | 455 | 4.3 | 5.7 | 10.5 |
| CBOW_DH_NS_11 | 7.82 | 78.95 | 0.513 | 4179 | 403 | 4.9 | 6.5 | 10.4 |
| CBOW_DH_NS_80 | 7.41 | 75.63 | 0.449 | 3969 | 403 | 5.2 | 6.4 | 9.8 |
| CBOW_DH_NS_160 | 6.23 | 66.47 | 0.368 | 3796 | 405 | 5.4 | 6.4 | 9.4 |
| CBOW_DHF | **7.32** | **78.75** | **0.557** | 4451 | 418 | 4.6 | 6.2 | 10.6 |
| CBOW_DHF_NS_11 | 4.79 | 73.60 | 0.548 | 3400 | 366 | 6.0 | 7.1 | 9.3 |
| CBOW_DHF_NS_80 | 5.50 | 66.22 | 0.458 | 2866 | 368 | 7.2 | 7.1 | 7.8 |
| CBOW_DHF_NS_160 | 5.68 | 60.92 | 0.384 | 3024 | 366 | 6.8 | 7.1 | 8.3 |
| CBOW_BPE_20K | 8.28 | 34.82 | 0.513 | 1371 | 103 | 15.0 | 25.2 | 13.3 |
| CBOW_BPE_40K | 8.96 | 40.66 | 0.493 | 1349 | 100 | 15.2 | 26.1 | 13.6 |
| CBOW_BPE_200K | 11.13 | 35.96 | 0.537 | 1343 | 96 | 15.3 | 27.0 | 14.0 |

Table 9: Standard deviation of the score obtained with word analogy task on QW. Lower is better.

| algorithmic variant | English | | German | | Russian | |
|---|---|---|---|---|---|---|
| | sem. | syn. | sem. | syn. | sem. | syn. |
| SG_original | 0.28 | 0.28 | 0.43 | 0.36 | 0.33 | 0.83 |
| *Our work:* | | | | | | |
| SG_code_opt | 0.76 | 0.55 | 1.28 | 0.76 | 0.39 | 1.24 |
| SG_no_subword | 1.10 | 0.76 | 0.96 | 0.66 | 0.55 | 1.82 |
| SG_batch | 0.51 | 0.33 | 1.54 | 0.52 | 0.40 | 0.98 |
| SG_NS_CT | 0.39 | 0.51 | 0.69 | 0.88 | 0.25 | 0.95 |
| SG_BPE_20K | 1.51 | 0.28 | 0.76 | 0.38 | 0.14 | 0.23 |
| SG_BPE_40K | 0.78 | 0.59 | 1.04 | 0.50 | 0.33 | 0.48 |
| SG_BPE_200K | 1.93 | 0.54 | 0.84 | 0.64 | 0.36 | 2.46 |
| CBOW_original | 0.13 | 0.23 | 0.10 | 0.21 | 0.08 | 0.63 |
| *Our work:* | | | | | | |
| CBOW_code_opt | 0.07 | 0.19 | 0.11 | 0.35 | 0.11 | 0.47 |
| CBOW_no_subword | 1.04 | 0.37 | 0.38 | 0.29 | 0.32 | 0.74 |
| CBOW_NS_11 | 0.04 | 0.48 | 0.18 | 0.48 | 0.08 | 0.57 |
| CBOW_NS_80 | 0.30 | 0.41 | 0.09 | 0.29 | 0.11 | 0.72 |
| CBOW_NS_160 | 0.13 | 0.29 | 0.11 | 0.37 | 0.10 | 0.47 |
| CBOW_DH | 0.34 | 0.33 | 0.11 | 0.40 | 0.12 | 0.44 |
| CBOW_DH_NS_11 | 0.10 | 0.40 | 0.17 | 0.39 | 0.21 | 1.25 |
| CBOW_DH_NS_80 | 0.30 | 0.38 | 0.20 | 0.42 | 0.24 | 0.53 |
| CBOW_DH_NS_160 | 0.26 | 0.57 | 0.06 | 0.25 | 0.17 | 0.91 |
| CBOW_DHF | 0.31 | 0.24 | 0.05 | 0.38 | 0.14 | 0.81 |
| CBOW_DHF_NS_11 | 0.13 | 0.75 | 0.13 | 0.63 | 0.17 | 1.35 |
| CBOW_DHF_NS_80 | 0.19 | 0.76 | 0.06 | 0.52 | 0.14 | 0.60 |
| CBOW_DHF_NS_160 | 0.12 | 0.55 | 0.07 | 0.23 | 0.14 | 1.06 |
| CBOW_BPE_20K | 0.27 | 0.25 | 0.24 | 0.38 | 0.23 | 0.69 |
| CBOW_BPE_40K | 0.45 | 0.37 | 0.48 | 0.31 | 0.41 | 0.46 |
| CBOW_BPE_200K | 0.24 | 0.28 | 0.46 | 0.14 | 0.10 | 1.28 |

