# OpenReview forum: "Faster Training of Word Embeddings"
_ICLR.cc/2021/Conference — Reject_

### Official Review · AnonReviewer2 · 2020-10-22
**Need technical depth**

**Rating:** 5
**Confidence:** 3

**Review:**

This paper proposes the approaches that reduce the training times of word2vec and fastText. To improve the efficiency, it uses several techniques such as negative sample sharing, batched updates, and a byte-pair encoding-based alternative for subword units. By using English, German, and Russian languages data, it shows that the proposed approach is much faster than the existing approaches.

Basically, I like the motivation and results of the paper; it is quite an important research problem to reduce training times of popular word embedding approaches. Besices, this paper is well structured and easy to follow; contributions are clear and proposed approaches are well described. However, the approaches proposed in the paper are somewhat technically shallow. I think more technical depth is needed for ICLR papers. This paper seems to be an engineering or industrial paper rather than a research paper; ICLR is not the best venue for the paper. To improve the paper, it is good to clearly describe the highlight of the proposed approach and explicitly show its theoretical property. While several approaches are described in Section 3, I think one or two approaches with detailed technical analyses are enough.

---

> ### Author Response · Authors · 2020-11-24
> **Response to Official Blind Review #2**
>
> > This is more an engineering paper: not a right / theoretically sufficient topic for ICLR.
>
> Reply: The performance optimization and the resulting library are major research contributions of the paper. **[Please refer to our general reply to the concern of “implementation paper.”](https://openreview.net/forum?id=v5WXtSXsVCJ&noteId=m2Wv8RJ4RE)**
>
> > More technical depth is needed: explain all the techniques including their theoretical properties. Only 2 approaches are explained well enough.
>
> Reply: In Section 3, we have added more details on how our code variants affect the execution. The techniques we adapt from other papers are evaluated only empirically by their original authors, with little to no discussion on theoretical properties: we have made an attempt to provide intuitions on these nevertheless.

---

### Official Review · AnonReviewer3 · 2020-10-25
**impressive speedup but not novel enough**

**Rating:** 5
**Confidence:** 3

**Review:**

This paper applies several algorithmic and code optimization techniques to reduce the training time for word2vec and fastText. The final implementation runs 3-20x times faster than the baselines on manycore CPUs, with almost no quality loss. The improved implementation is a result of the combination of many techniques, including namely no_subword, Minibatching, Negative sharing etc.


*Strong points*
- The 3-20x speedup is impressive.
- Given the popularity of word2vec and fastText, the open-sourced version of the proposed method might benefit many applications.
The experiment is solid.

*Weak points*
- This paper is more like a technical report, describing the details of an improved implementation. Many of the optimization techniques being used can also be found in previous works. Therefore the novelty of this paper might be slightly below the standard of ICLR.
- This paper is hard to follow, partly because of the organization of this paper. It would be beneficial to address the following potential issues,
     +  Two many engineering details are included. Consider moving some of those details into appendix, and include more high level intuition why those optimization should be included.
     + Better to include an overview of techniques before explaining the detailed implementation. This helps the audience quickly understand the major contributions/novelties compared with existing works.
     + The main text frequently refers to the line# of the source code, which hurts the readability of this paper.
Ideally, provide a table to summarize the datasets being used in this paper.

---

> ### Author Response · Authors · 2020-11-24
> **Response to Official Blind Review #3**
>
> > More like a technical report, describing the details of an improved implementation / Too many engineering details, maybe move them to the appendix?
>
> Reply: The careful performance optimization including the techniques applied and the resulting library are the major research contribution of the paper, which is why we describe it in some detail. **[Please see our general reply to the concern of “implementation paper.”](https://openreview.net/forum?id=v5WXtSXsVCJ&noteId=m2Wv8RJ4RE)**
>
> > The paper would be more readable with an overview of techniques presented before the individual techniques are described.
>
> Reply: We have added a short overview in the beginning of Section 3.
>
> > References to the lines of code in the text make it hard to read.
>
> Reply: We prefer to keep these references because we believe they help with the exact explanation of what and how we improved performance and because the other reviewers commented that the paper is easy to read.

---

### Official Review · AnonReviewer1 · 2020-10-28
**Official Blind Review #1**

**Rating:** 4
**Confidence:** 4

**Review:**

The paper describes a few implementation tricks to improve the performance of word2vec and fastText algorithms on CPUs. The authors offer an implementation that is optimized with intrinsics. In addition to that, the authors discuss a few different implementational improvements that can be made at the algorithmic level to improve the training speed of fastText.

Pros:
- The paper is easy to read and follow
- The authors have tried a variety of implementation techniques aimed at improving training performance
- The experimental results shows large speedups on CPUs in a few cases without trading off accuracy.

Cons:
The novelty as well as the importance of this work is not clear. The code optimizations mentioned in the paper using AVX-512 intrinsics targeted at Intel CPUs, blocking and merging, or load/store optimizations of hidden layer and gradients aren't particularly new and neither are they specific to fastText. The algorithmic implementation variants tried on the other hand such as ignoring subwords or batching for skip-grams are specific to fastText, but are fairly straight-forward. The only novel contributions to the algorithm that result in improvements appear to be dynamic hidden-layer update and byte-pair encoding. Additionally, the paper fails to motivate why this line of optimization is important while there appear to be other straight-forward ways to make the training significantly faster (distributed training, clean GPU implementation that is open-source, etc).

---

> ### Author Response · Authors · 2020-11-24
> **Response to Official Blind Review #1**
>
> > Only Dynamic Hidden and usage of BPE tokens are novel contributions that bring benefit.
>
> Reply: Another major contribution is the carefully optimized implementation of our multi-algorithmic approach as an open source library. **[Please refer to our general reply to the concern of “implementation paper.”](https://openreview.net/forum?id=v5WXtSXsVCJ&noteId=m2Wv8RJ4RE)**
>
> > Why not use a GPU or distributed computing instead?
>
> Reply: We explain in the paper that the current state of the art GPU implementation (BlazingText) can only achieve the performance of a 16-threaded baseline implementation. From their results we can derive that our best multi-threaded code would improve over a single GPU implementation by 4-20x depending on the choice of Skipgram/CBOW and subwords/no subwords.
>
> What is more, we have now compared 20-threaded runs of our no_subword variant to the 20-threaded runs of the original word2vec code (which was not included in the paper), and achieved 5-6x speedups. This is much greater than those obtained on a K20 GPU by Bae & Yi (2016) for word2vec with negative sampling over (only) 12-threaded CPU runs (1.4-1.6x speedups). We’ve added a discussion on this in the related work.
>
> Indeed, not every algorithm is suitable for GPUs if it lacks the very fine-grained parallelism for SIMD multithreading. Fine-grained parallelism is limited partially by small vector sizes typically used to represent word embeddings. In addition to that, word2vec and fastText are memory-intensive algorithms where the potential benefits of parallelism are capped by memory bandwidth.
>
> A large-scale distributed implementation is expected to yield further speedups: we have added a short comment in the section on related work, on why we do not need more communication than required by the original fastText algorithm. Furthermore, our contribution would still apply for the computation inside the nodes. However, distributed computing requires a cluster. In many real-world settings a cluster may not be available (e.g., an embedded scenario) or the users may just want to run training on their own workstation.

---

### Official Review · AnonReviewer4 · 2020-10-28
**Official Blind Review**

**Rating:** 7
**Confidence:** 3

**Review:**

The current work introduces a new approach that speeds up the training process of word embedding models such as word2vec and fasttext on specific hardware.
The improvements are enumerated as follows: Code performance optimizations, using AVX-512 extensions, suppression of subword information (only for fasttext), modifying batch over training, Negative sampling shared for consecutive words, Dynamically updating the hidden layer and the use of Byte-Pair vocabulary.
Tests are conducted on a set of word based benchmarks in many languages, where improvements in terms of both performance and training time is shown, being CBOW_no_subword the best configuration in terms of speed up across languages.

By and large the current work is ok. The authors presented in a clear way what is the current state, the models they build on top of, the techniques used and present experimentation that show the effectiveness of such improvements.
My main concern is, on the experimentation, where I would have chosen more standard datasets. I guess WordSim 353 is a must for a semantic similarity evaluation and its multilingual extension. From what I saw in the appendix, MUSE is an average of many datasets, which includes WS353.
In addition, a discussion or application of the author's improvements on GloVe seems also reasonable.
Despite that, I think the submission is strong.
Some minor recommendations below.

Evaluation: Averaging on several monolingual datasets is ok, if you present the individual scores as well. In fact, columns showing almost no difference between models. I would recommend to include some individual starts, for instance WS353, SimLex999, which have also multilingual counterparts from Leviant & Reichart (2015)

Table 1/2/3: Include a comment (or an arrow) to clarify if the scores are higher (or lower) the better.

---

> ### Author Response · Authors · 2020-11-24
> **Response to Official Blind Review #4**
>
> > We should list individual scores (e.g., WS353, SimLex999) for MUSE:
>
> Reply: We have added these scores as tables to Appendix B. Note that the results are based on embeddings obtained from different runs than those in the original paper, so the scores slightly vary from those presented in the Evaluation section.
>
> > Scores in tables: “higher is better” / “lower is better”.
>
> Reply: We have added clarification comments to the table captions in the main paper and the Appendix.
>
> > “In addition, a discussion or application of the author's improvements on GloVe seems also reasonable. Despite that (...)”
>
> Reply: GloVe uses a different mechanism and different data structures than word2vec and fastText. In particular, GloVe cannot be trained in a streaming fashion since the full co-occurrence matrix needs to be pre-computed. Therefore, our optimizations cannot be straightforward applicable. Even code_opt would require different optimization techniques, as some of those we use are specific to the structure of the shallow neural network fastText and word2vec are built upon. We have added a relevant paragraph to the related work.

---

### Author Response · Authors · 2020-11-24
**Reply to all reviewers**

We would like to thank all reviewers for their time and effort. Three of the four reviewers had a common concern that we would like to address here jointly.

> Code optimization techniques are not new / paper is more like a technical report describing an optimized implementation / too many engineering details / paper is more an engineering or industrial rather than research paper

Reply: A major (but not the entire) contribution of the paper is indeed a carefully optimized implementation for state-of-the-art manycore CPUs. How to do this in a careful, architecture-cognizant way is not just engineering but research. For example in the well-established area of scientific computing, but also in embedded computing, entire conferences are dedicated to this problem of how to create highly optimized implementations and the areas benefit from the availability of the corresponding available open source implementations and from the knowledge of how to create them. The available high level methods (SIMD vectorization, blocking for locality, loop merging, scalar replacement, establishing instruction-level parallelism etc., parallelization, avoiding false sharing, and many more) are well-understood but instantiate themselves very differently for different algorithms. How to do this is the research and the measure of success is the speedup obtained. **We believe that ICLR explicitly values and asks for this type of research by explicitly listing in the Call for Papers at https://iclr.cc/Conferences/2021/CallForPapers among the relevant topics: “implementation issues, parallelization, software platforms, hardware.”**

---

### Decision · Program_Chairs · 2021-01-07
**Final Decision**

**Decision:**

Reject

**Comment:**

This work describes a series of strategies for optimizing the training speed of
word embeddings (as in word2vec and fasttext).

All reviewers appreciate the convincing empirical results, which are without a
doubt impressive.  Reviewers also mostly agree that speeding up embedding
training is important, and there is no doubt that this type of paper is
appropriate for ICLR (as clearly highlighted in the CfP.)

However, the specific optimization strategies deployed and described here
are deemed not to bring novel insight, useful in itself to the community, beyond the
software contribution described.
The paper seems to mostly serve as documentation of the
implementation, limiting its value and impact to further research.
The pedagogic value is also limited, as the paper tackles multiple different,
eclectic optimizations, a narrative strategy that does not leave room to describe a single one more
generally, helping the community find other places to apply it.
All in all this leads to a borderline negative assessment, and I cannot
recommend acceptance.